

# Association between *Vitamin D receptor (VDR)* gene polymorphisms and hypertensive disorders of pregnancy: a systematic review and meta-analysis

Yicong Guo[1,*], Yu Zhang[2,*], Xiangling Tang[2], Xionghao Liu[2] and Huilan Xu[1]

[1] Department of Social Medicine and Health Management, Xiangya School of Public Health, Central South University, Changsha, Hunan, China
[2] Center for Medical Genetics & Hunan Key Laboratory of Medical Genetics, School of Life Sciences, Central South University, Changsha, Hunan, China
[*] These authors contributed equally to this work.

Corresponding authors
Xionghao Liu, liuxionghao@sklmg.edu.cn
Huilan Xu, xhl_csu@163.com

## ABSTRACT

**Background**. Hypertensive disorders of pregnancy (HDP) are currently one of the major causes of pregnancy-related maternal and fetal morbidity and mortality worldwide. Recent studies provide evidence that maternal *Vitamin D receptor (VDR)* gene polymorphisms probably play a key role by affecting the biological function of vitamin D in some adverse pregnancy outcomes, while the relationship between the *VDR* gene polymorphisms and the risk of HDP remains controversial in current studies. This systematic review and meta-analysis aimed to comprehensively evaluate the association of the *VDR* gene polymorphisms with HDP susceptibility.

**Methods**. This meta-analysis follows the Preferred Reporting Items for Systematic Reviews and Meta-Analyses (PRISMA) statement and a protocol has been registered in the PROSPERO (ID: CRD42022344383) before commencing this review. PubMed, Web of Science, Embase, and the Cochrane Library databases were searched until January 21, 2023. Case-control and cohort studies that reported the association of the *VDR* gene polymorphisms with HDP were included. The quality of the included studies was assessed using the Newcastle-Ottawa Scale (NOS) for non-randomized studies. The odds ratios (ORs) with corresponding 95% confidence intervals (CIs) of the five models (allele model, dominant model, recessive model, homozygous model, heterozygous model) were pooled respectively, and subgroup analysis was performed based on ethnicity.

**Results**. A total of ten studies were included. The *VDR* gene *ApaI* polymorphism was associated with HDP susceptibility in the dominant model (OR: 1.38; 95% CI [1.07–1.79]; $P = 0.014$) and the heterozygote model (OR: 1.48; 95% CI [1.12–1.95]; $P = 0.006$). In subgroup analysis, the heterozygote model (OR: 2.06; 95% CI [1.21–3.52]; $P = 0.008$) of the *ApaI* polymorphism was associated with HDP in Asians, but not in Caucasians.

**Conclusion**. The *VDR* gene *ApaI* polymorphism may be associated with HDP susceptibility. Insufficient evidence to support the existence of ethnic differences in this association.

## INTRODUCTION

Hypertensive disorders of pregnancy (HDP) are mainly characterized by persistently elevated blood pressure (BP) levels equal to or more than 140/90 mmHg and the resulting pathological changes, typically encompassing the following four categories: chronic hypertension (occurring before 20 weeks' gestation or persisting longer than 12 weeks after delivery), gestational hypertension (occurring after 20 weeks' gestation), preeclampsia, or preeclampsia superimposed on chronic hypertension (*Metoki et al., 2022*). HDP are currently one of the major causes of pregnancy-related maternal and fetal morbidity and mortality worldwide (*Roberts et al., 2013*). The prevalence of HDP, gestational hypertension, and preeclampsia ranges respectively from 5.2 to 8.2%, 1.8 to 4.4%, and 0.2 to 9.2% in various regions of the world (*Umesawa & Kobashi, 2017*). In Latin America and the Caribbean, hypertensive disorders are responsible for almost 26.0% of maternal deaths, whereas in Africa and Asia they contribute to 9.0% of deaths (*Khan et al., 2006*; *Steegers et al., 2010*). Multiple risk factors contribute to the onset of HDP, and some of them are widely recognized, such as maternal age, obesity, smoking, alcohol intake, gestational diabetes mellitus (GDM), *etc.* (*Antza, Cifkova & Kotsis, 2018*; *Bartsch et al., 2016*). In addition, current evidence suggested that some maternal genetic variants may also play a significant role in the development of HDP (*Umesawa & Kobashi, 2017*), including the *angiotensin-converting enzyme* (*ACE*) gene (*Dmitrenko et al., 2020*), *angiotensinogen (AGT)* gene (*Zhu et al., 2012*), *endothelial nitric oxide synthase* (*eNOS*) gene (*Alpoim et al., 2014*), *methylenetetrahydrofolate reductase* (*MTHFR*) gene (*Xia, Chang & Cao, 2012*), *tumor necrosis factor-α* (*TNF-α*) gene (*Lin et al., 2019*), *catechol-O-methyltransferase* (*COMT*) gene (*Taravati et al., 2017*), which were commonly considered candidate genes for prediction.

Vitamin D status has been considered another important, modifiable nutrition-related risk factor for HDP in recent studies (*Bodnar et al., 2014*; *Tabesh et al., 2013*). Epidemiologic investigations indicated that vitamin D deficiency or blocked utilization was associated with the increased risk of HDP (*Kiely et al., 2016*; *Serrano et al., 2018*), and calcium and vitamin D supplementation were confirmed to decrease the risk of preeclampsia when compared to placebo by several meta-analyses (*Fogacci et al., 2020*; *Khaing et al., 2017*; *Morales-Suárez-Varela et al., 2022*; *Palacios et al., 2016*). 1,25-Dihydroxyvitamin $D_3$ (1,25-$(OH)_2D_3$), as the active form of vitamin D, mediates its physiological effects by specific interactions with the vitamin D receptor (VDR). The VDR is a DNA-binding transcription factor that is a member of the steroid receptor family in the cell nucleus. When specifically binds to 1,25-Dihydroxyvitamin $D_3$, VDR generates an active signal transduction complex consisting of a heterodimer of the 1,25-$(OH)_2D_3$-liganded VDR and unoccupied retinoid X receptor (RXR). This liganded VDR-RXR heterodimer can recognize vitamin D responsive elements (VDREs) in the DNA sequence of vitamin D-regulated genes (*Haussler et al., 2011*;

*Haussler & Norman, 1969*; *Jin et al., 1996*). Genetically, VDR is encoded by the *VDR* gene located in 12q13.11 on the chromosome, which consists of two promoter regions, eight coding exons (namely, 2-9), and six untranslated exons (1A-1F) (Fig. S1) (*Jehan, d'Alésio & Garabédian, 2007*; *Uitterlinden et al., 2004*; *Valdivielso & Fernandez, 2006*). Polymorphisms of the *VDR* gene have been shown to alter VDR functions that affect vitamin D activities and metabolic concentrations (*Maestro et al., 2016*). Four common single nucleotide polymorphisms (SNPs) of the *VDR* gene are most intensively studied, including the *ApaI* polymorphism (rs7975232), the *BsmI* polymorphism (rs1544410), the *FokI* polymorphism (rs2228570, also known as rs10735810) and the *TaqI* polymorphism (rs731236). Among the four SNPs, three of them occur in the intron sections (the *TaqI*, *ApaI*, and *BsmI* variants), while only the *FokI* variant changes the codon (Fig. S1) (*Haussler et al., 1997*). Nevertheless, each polymorphism of the VDR can exert different effects, for instance, the *BsmI* and *TaqI* polymorphisms do not modify the VDR protein structure, but they can influence the stability and/or translation efficiency of the RNA (*Jurutka et al., 1997*).

Although previous meta-analyses have found that the *VDR* gene polymorphisms could increase the susceptibility to essential hypertension (EH) (*Nunes et al., 2020*; *Zhu et al., 2019*), and the *VDR* gene polymorphisms were reported to be associated with plasma renin activity (*Vaidya et al., 2011*), the relationship between the *VDR* gene polymorphisms and the risk of HDP remains controversial in current studies. The results from current studies are inconsistent between populations from different regions or of different ethnicities. For example, *Farajian-Mashhadi et al. (2020)* reported that the maternal *VDR* gene *FokI* variant was associated with a decreased risk of preeclampsia; in contrast, one study conducted by *Zhan et al. (2015)* indicated that the G allele of the *FokI* polymorphism (A>G) increased the risk of preeclampsia among the Chinese population, while another study conducted in China showed that the association of the *FokI* polymorphism (A>G) with HDP susceptibility was not statistically significant (*Si et al., 2022*). In fact, this association has only been intensively investigated in recent years, and there has been no meta-analysis published assessing the association comprehensively. Therefore, we conducted this meta-analysis to investigate the association of the *VDR* polymorphisms with HDP susceptibility.

## MATERIAL AND METHODS

A protocol was registered before commencing this review in the International Prospective Register of Systematic Reviews PROSPERO (ID: CRD42022344383). The current meta-analysis follows the Preferred Reporting Items for Systematic Reviews and Meta-Analyses (PRISMA) statement (*Moher et al., 2010*). The PRISMA checklist for reporting the meta-analysis was shown in Table S1.

### Search strategy

Original articles from PubMed, Web of Science, EMBASE, and the Cochrane Library databases were systematically searched from the founding date of each database to January 21, 2023. A combination of the following searching terms was used: ("VDR" OR "vitamin D receptor" OR "FokI" OR "rs2228570" OR "BsmI" OR "rs1544410" OR "ApaI" OR "rs7975232" OR "TaqI" OR "rs731236") AND ("polymorphisms" OR
"SNPs" OR "genotype" OR "variant" OR "mutation") AND ("hypertensive disorders of pregnancy" OR "gestational hypertension" OR "gestational hypertensive disorders" OR "pre-eclampsia"). The search strategies for each database are detailed in Table S2. In addition, we also screened the references of relevant articles to identify additional published and unpublished records. Yu Zhang and Yicong Guo performed the search strategy. The disagreement was settled by a third reviewer's (Xiangling Tang) evaluation and discussed until a consistent result was reached.

## Inclusion and exclusion criteria

The studies which met the following explicit criteria were included: (1) case-control or cohort design; (2) the relationship between the *VDR* gene polymorphisms and the risk of HDP was reported; (3) providing sufficient data about the genotype frequencies of the *VDR* gene polymorphisms for calculating the value of odds ratio (OR) and 95% confidence interval (CI); (4) the distribution of genotypes of controls were in accordance with the Hardy-Weinberg equilibrium (HWE); (5) studies were published or written in English.

The exclusion criteria were: (1) reviews, case reports, letters, conference abstracts, and comments; (2) *in vivo* or *in vitro* experiments; (3) studies containing overlapping or insufficient data; (4) duplicate studies retrieved from various databases.

## Data extraction and quality assessment

The following information from eligible studies was extracted or calculated based on genotype distribution: (1) the first author's name, publication year, country, ethnicity, genotyping methods, types of HDP, and the *VDR* gene variants; (2) sample size, age, and genotype distribution in both case and control groups; (3) odds ratios (ORs) and corresponding 95% confidence intervals (CIs); (4) the HWE test results for the control group. All data were extracted independently by two researchers (Yu Zhang and Yicong Guo), and if there were disagreements, questions were discussed and resolved by a third reviewer (Xiangling Tang).

The quality of the included studies was assessed using the Newcastle-Ottawa Scale (NOS) for non-randomized studies. The NOS is a rating scale in which points are awarded to studies based on selection, comparability, and exposure or outcome, where each study score ranges from 0 to 9 points (*Stang, 2010*). A study with a total quality score of more than seven points was considered a high-quality study. Two reviewers (Yu Zhang and Yicong Guo) independently rated the quality of the included studies, and the differences in ratings between reviewers were also resolved by discussion with a third reviewer (Xiangling Tang).

## Statistical analysis

The HWE of genotypes in each control group was determined using the Chi-square test. The pooled ORs and corresponding 95% CIs of the five models (allele model, homozygous model, heterozygous model, dominant model, and recessive model) were calculated respectively, to evaluate the association between the *VDR* gene polymorphisms (*ApaI*, *BsmI*, *TaqI*, and *FokI*) and the risk of HDP. The heterogeneity was evaluated by Cochran's Q-statistic test and I-squared ($I^2$) (*Chen & Benedetti, 2017*; *Higgins et al., 2003*). If $I^2$>50%

and $P < 0.10$, the random effect model was used, otherwise the fixed effect model was applied (*DerSimonian & Laird, 1986*). Subgroup analysis grouped by ethnicity (Caucasian and Asian) was performed to investigate the ethnic differences of this association. Sensitivity analysis was performed to evaluate the effect of a particular study on the overall results by deleting one study at a time and combining the effect values of the remaining studies. In addition, we assessed the publication bias by Egger's test (*Hayashino, Noguchi & Fukui, 2005*) and Begg's test (*Begg & Mazumdar, 1994*), and a visualized funnel plot was performed as a complement.

All statistical analyses were performed using Stata v16.0 (Stata Corp LP, College Station, TX, USA). A two-sided $P < 0.05$ was considered statistically significant except for Cochran's Q test. In our study, all analyses were based on previously published research; thus, no ethical approval or patient consent was required.

# RESULTS

## Study selection

Figure 1 provided the flowchart of the literature search process. Our study yielded 177 potentially relevant articles in four electronic databases: 45 from PubMed, 67 from Embase, 63 from Web of Science, and two from the Cochrane Library. After excluding duplicate studies, 143 articles were retained. Of the 143 studies initially identified, 119 were excluded because they failed to meet the inclusion criteria based on title and abstract review. The full texts of the remaining 24 articles were reviewed for eligibility, and 14 articles were excluded for various reasons, including comments ($n = 2$), the *VDR* gene polymorphisms were not measured ($n = 8$), and other outcomes ($n = 4$). We finally selected a total of ten qualified articles (*Aziz et al., 2022*; *Caccamo et al., 2020*; *Farajian-Mashhadi et al., 2020*; *Ghorbani et al., 2021*; *Magiełda-Stola et al., 2021*; *Rezavand et al., 2019*; *Rezende et al., 2012*; *Setiarsih, Hastuti & Nurdiati, 2022*; *Si et al., 2022*; *Zhan et al., 2015*), including 1,558 cases and 5,119 controls in the meta-analysis.

## Characteristics and quality of the included studies

The characteristics and genotype frequencies of all the included studies were summarized in Tables 1 and 2. Among the ten studies, six studies (*Aziz et al., 2022*; *Farajian-Mashhadi et al., 2020*; *Ghorbani et al., 2021*; *Magiełda-Stola et al., 2021*; *Rezende et al., 2012*; *Si et al., 2022*) were analyzed for the *ApaI* polymorphism, six studies (*Caccamo et al., 2020*; *Farajian-Mashhadi et al., 2020*; *Magiełda-Stola et al., 2021*; *Rezavand et al., 2019*; *Rezende et al., 2012*; *Zhan et al., 2015*) for the *BsmI* polymorphism, eight studies (*Caccamo et al., 2020*; *Farajian-Mashhadi et al., 2020*; *Magiełda-Stola et al., 2021*; *Rezavand et al., 2019*; *Rezende et al., 2012*; *Setiarsih, Hastuti & Nurdiati, 2022*; *Si et al., 2022*; *Zhan et al., 2015*) for the *FokI* polymorphism and four studies (*Farajian-Mashhadi et al., 2020*; *Magiełda-Stola et al., 2021*; *Rezavand et al., 2019*; *Setiarsih, Hastuti & Nurdiati, 2022*) for the *TaqI* polymorphism. Regarding the subjects' ethnicity, there were seven studies (*Aziz et al., 2022*; *Farajian-Mashhadi et al., 2020*; *Ghorbani et al., 2021*; *Rezavand et al., 2019*; *Setiarsih, Hastuti & Nurdiati, 2022*; *Si et al., 2022*; *Zhan et al., 2015*) on Asians and three studies (*Caccamo et al., 2020*; *Magiełda-Stola et al., 2021*; *Rezende et al., 2012*) on Caucasians. Six studies

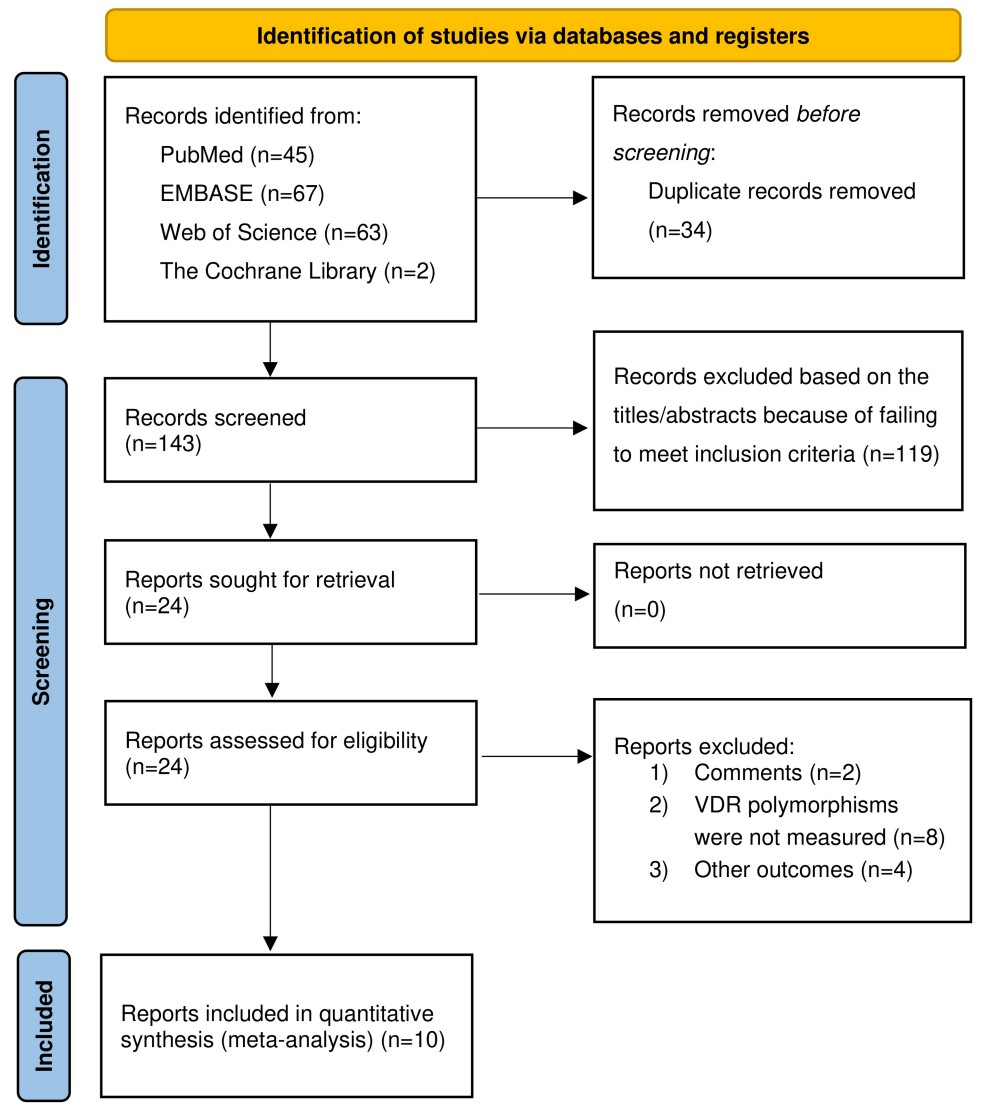

**Figure 1** Flow chart of the included studies of meta-analysis.

(*Aziz et al., 2022*; *Farajian-Mashhadi et al., 2020*; *Ghorbani et al., 2021*; *Magiełda-Stola et al., 2021*; *Rezavand et al., 2019*; *Zhan et al., 2015*) included only patients with preeclampsia as case groups, and patients with both gestational hypertension and preeclampsia were involved in the remaining four studies (*Caccamo et al., 2020*; *Rezende et al., 2012*; *Setiarsih, Hastuti & Nurdiati, 2022*; *Si et al., 2022*). The distribution of genotypes in controls was not completely in accordance with HWE in two studies (*Farajian-Mashhadi et al., 2020*; *Rezavand et al., 2019*), thus they were excluded in the subsequent meta-analysis. Of the ten studies included, four studies (*Caccamo et al., 2020*; *Magiełda-Stola et al., 2021*; *Si et al., 2022*; *Zhan et al., 2015*) scored 7 or higher and were considered high quality, five studies (*Farajian-Mashhadi et al., 2020*; *Ghorbani et al., 2021*; *Rezende et al., 2012*; *Setiarsih,*

*Hastuti & Nurdiati, 2022*) were rated 6, and one study (*Rezavand et al., 2019*) with a score of 5, indicating that the overall quality was acceptable (Table S3).

### *VDR* gene polymorphisms and the risk of HDP

Table 3 showed the pooled results of the four SNPs based on the five models. For the *VDR* gene *ApaI* polymorphism, statistically significant associations with HDP susceptibility were found in the overall population in the dominant model (aa + Aa *vs.* AA: OR: 1.38; 95% CI [1.07–1.79]; $P = 0.014$) (Fig. 2A) and the heterozygote model (Aa *vs.* AA: OR: 1.48; 95% CI [1.12–1.95]; $P = 0.006$) (Fig. 2B). Subgroup analysis based on ethnicity showed that the heterozygote model (Aa *vs.* AA: OR: 2.06; 95% CI [1.21–3.52]; $P = 0.008$) of the *ApaI* polymorphism was associated with an increased risk of HDP in Asians but not in Caucasians.

A statistically significant association was observed between the *VDR* gene *Bmsl* polymorphism and the risk of HDP in the overall population in the homozygote model (bb *vs.* BB: OR: 0.72; 95% CI [0.56–0.99]; $P = 0.042$) (Fig. 3). Besides, no statistically significant associations were found between the *BsmI* polymorphism and HDP when stratified by ethnicity.

The *VDR* gene *FokI* polymorphism was only found statistically associated with the risk of HDP in Caucasians based on the recessive model (ff *vs.* Ff + FF: OR: 1.43; 95% CI [1.01–2.03] $P = 0.041$) (Fig. 4A). In the overall population, no statistically significant associations were observed between the *FokI* polymorphism and HDP in the recessive model (ff *vs.* Ff + FF: OR: 1.23; 95% CI [0.88–1.73]; $P = 0.228$) (Fig. 4B).

The *VDR* gene *TaqI* polymorphism had no significant associations with the risk of HDP in both the overall and Asian populations according to the five models. In addition, in subgroup analysis, only one study investigated this relationship among Caucasians and reported a statistically significant association between the *TaqI* polymorphism and HDP susceptibility in the allele model (t *vs.* T: OR: 1.42; 95% CI [1.02–1.98] $P = 0.040$).

### Sensitivity analyses and publication bias

Sensitivity analyses were conducted by removing each study included from the meta-analysis at a time. After the included studies were successively removed, the estimates were statistically significant with OR ranging from 1.36 (95% CI [1.00–1.85]) to 1.72 (95% CI [1.18–2.50]) in the dominant model (aa + Aa *vs.* AA) (Fig. 5A) and from 1.39 (95% CI [1.04–1.87]) to 1.77 (95% CI [1.21–2.60]) in the heterozygote model (Aa *vs.* AA) (Fig. 5B), indicating that the overall results were relatively stable. Begg's test and Egger's test did not show any evidence of publication bias among the included studies (Table 3), and the Egger funnel plots of the results of the included studies were approximately symmetrical (Figs. S2–S6).

### DISCUSSION

To provide a better understanding of the relationship between the *VDR* gene polymorphisms and HDP susceptibility, we conducted this systematic review and meta-analysis. As far as we know, this is the first meta-analysis to comprehensively investigate

Guo et al. (2023), *PeerJ*, DOI 10.7717/peerj.15181

**Table 1  Characteristics of included studies in the meta-analysis.**

| Authors | Year | Country | Ethnicity | Disease | Age, y | | Sample size, n | | Genotyping methods | SNPs | NOS |
|---|---|---|---|---|---|---|---|---|---|---|---|
| | | | | | Case | Control | Case | Control | | | |
| *Rezende et al.* | 2012 | Brazil | Caucasian | GH, PE | 28.1 ± 6.8 | 26.6 ± 6.1 | 316 | 213 | PCR-RFLP | *ApaI, BsmI, FokI* | 6 |
| *Zhan et al.* | 2015 | China | Asian | PE | 30.7 ± 5.7 | 30.7 ± 4.5 | 402 | 554 | TaqMan qPCR | *BsmI, FokI* | 7 |
| *Rezavand et al.* | 2019 | Iran | Asian | PE | 31.4 ± 6.4 | 29.0 ± 6.0 | 100 | 100 | PCR-RFLP | *BsmI, FokI, TaqI* | 5 |
| *Caccamo et al.* | 2020 | Italy | Caucasian | GH, PE | 33.0 ± 6.2 | 33.0 ± 5.9 | 116 | 69 | TaqMan qPCR | *BsmI, FokI* | 7 |
| *Farajian-Mashhadi et al.* | 2020 | Iran | Asian | PE | 27.6 ± 6.4 | 28.1 ± 6.4 | 152 | 160 | PCR-RFLP | *ApaI, BsmI, FokI, TaqI* | 6 |
| *Magiełda-Stola et al.* | 2021 | Poland | Caucasian | PE | 30.1 ± 5.5 | 30.6 ± 4.4 | 122 | 184 | PCR-RFLP | *ApaI, BsmI, FokI, TaqI* | 7 |
| *Ghorbani et al.* | 2021 | Iran | Asian | PE | 31.4 ± 6.4 | 29.0 ± 6.0 | 100 | 100 | PCR-RFLP | *ApaI* | 6 |
| *Si et al.* | 2022 | China | Asian | GH, PE | 29.3 ± 4.0 | 28.7 ± 3.6 | 105 | 3594 | MALDI-TOF MS PCR | *ApaI, FokI* | 8 |
| *Setiarsih, Hastuti & Nurdiati* | 2022 | Indonesia | Asian | GH, PE | 29.1 ± 6.9 | 27.1 ± 6.0 | 105 | 105 | PCR-RFLP | *FokI, TaqI* | 6 |
| *Aziz et al.* | 2022 | Pakistan | Asian | PE | 33.1 ± 5.3 | 27.6 ± 4.9 | 40 | 40 | AS-PCR | *ApaI* | 6 |

Notes.

GH, gestational hypertension; PE, pre-eclampsia; PCR-RFLP, polymerase chain reaction-restriction fragment length polymorphism; TaqMan qPCR, TaqMan-Based real-time polymerase chain reaction; MALDI-TOF MS PCR, matrix-assisted laser desorption ionization time-of-flight mass spectrometry coupled with single-base extension polymerase chain reaction; AS-PCR, allele-specific polymerase chain reaction; SNP, single nucleotide polymorphism; NOS, Newcastle-Ottawa Scale.
**Table 2   Genotype frequencies of vitamin D receptor gene polymorphisms in HDP patients and matched controls.**

| SNP | Authors | Genotype | | | | | | HWE *P* value |
|---|---|---|---|---|---|---|---|---|
| | | Case | | | Control | | | |
| *ApaI* (rs7975232) | | AA | Aa | aa | AA | Aa | aa | |
| | *Rezende et al.* | 92 | 156 | 68 | 70 | 98 | 45 | 0.329 |
| | *Farajian-Mashhadi et al.* | 36 | 95 | 21 | 45 | 91 | 24 | 0.046 |
| | *Magiełda-Stola et al.* | 38 | 61 | 23 | 40 | 97 | 47 | 0.449 |
| | *Ghorbani et al.* | 9 | 62 | 29 | 17 | 46 | 37 | 0.677 |
| | *Si et al.* | 11 | 15 | 3 | 371 | 270 | 57 | 0.427 |
| | *Aziz et al.* | 11 | 19 | 10 | 9 | 24 | 7 | 0.190 |
| *BsmI* (rs1544410) | | BB | Bb | bb | BB | Bb | bb | |
| | *Rezende et al.* | 52 | 159 | 105 | 36 | 107 | 70 | 0.651 |
| | *Zhan et al.* | 313 | 84 | 5 | 456 | 89 | 9 | 0.062 |
| | *Rezavand et al.* | 20 | 72 | 8 | 28 | 65 | 7 | <0.001 |
| | *Caccamo et al.* | 23 | 65 | 28 | 11 | 36 | 22 | 0.557 |
| | *Farajian-Mashhadi et al.* | 39 | 86 | 27 | 40 | 90 | 30 | 0.102 |
| | *Magiełda-Stola et al.* | 41 | 48 | 33 | 82 | 74 | 28 | 0.104 |
| *FokI* (rs2228570) | | FF | Ff | ff | FF | Ff | ff | |
| | *Rezende et al.* | 121 | 145 | 50 | 90 | 104 | 19 | 0.150 |
| | *Zhan et al.* | 63 | 176 | 163 | 101 | 292 | 161 | 0.117 |
| | *Rezavand et al.* | 6 | 22 | 72 | 7 | 38 | 55 | 0.900 |
| | *Caccamo et al.* | 55 | 43 | 18 | 31 | 27 | 11 | 0.227 |
| | *Farajian-Mashhadi et al.* | 106 | 38 | 8 | 89 | 54 | 17 | 0.052 |
| | *Magiełda-Stola et al.* | 30 | 58 | 34 | 40 | 102 | 42 | 0.140 |
| | *Si et al.* | 3 | 15 | 10 | 145 | 349 | 202 | 0.799 |
| | *Setiarsih, Hastuti & Nurdiati* | 16 | 53 | 36 | 7 | 50 | 48 | 0.205 |
| *TaqI* (rs731236) | | TT | Tt | tt | TT | Tt | tt | |
| | *Rezavand et al.* | 40 | 51 | 9 | 40 | 55 | 5 | 0.011 |
| | *Farajian-Mashhadi et al.* | 59 | 71 | 22 | 65 | 70 | 25 | 0.399 |
| | *Magiełda-Stola et al.* | 42 | 59 | 21 | 84 | 78 | 22 | 0.554 |
| | *Setiarsih, Hastuti & Nurdiati* | 98 | 7 | 0 | 97 | 8 | 0 | 0.685 |

**Notes.**

HDP, Hypertensive Disorders of Pregnancy; HWE, Hardy–Weinberg equilibrium;; SNP, single nucleotide polymorphism.

the associations between the four common SNPs of the *VDR* gene and HDP susceptibility by pooling ORs and the corresponding 95% CIs. Our study contributed to identifying the *VDR* gene as an additional candidate gene for subsequent Genome-Wide Association Studies (GWAS) for predicting HDP. Moreover, since genetic genes are rarely affected by environmental factors, spotting SNP loci associated with HDP provided potential instrumental variables for future Mendelian randomization (MR) design, which can reveal the association between various exposure and HDP while avoiding confounding factors and reverse causality (*Davey Smith & Hemani, 2014*; *Haycock et al., 2016*).

The results of our meta-analysis showed that the *VDR* gene *ApaI* polymorphism was associated with HDP susceptibility in the overall population without heterogeneity, especially in Asian populations. Pregnant women with the *ApaI* Aa polymorphism had

Guo et al. (2023), *PeerJ*, DOI 10.7717/peerj.15181

**Table 3  Meta-analysis of associations between the VDR gene polymorphisms and HDP.** Meta-analysis of associations between the VDR ApaI (rs7975232), BsmI (rs1544410), FokI (rs2228570) and TaqI (rs731236) polymorphisms and HDP.

| SNP | Comparison | Subgroup | No. of studies | Test of association | | | Tests of heterogeneity | | | Begg test for publication test | | Egger test for publication bias | |
|---|---|---|---|---|---|---|---|---|---|---|---|---|---|
| | | | | OR | 95% CI | *P*-value | Model | Q | *P*-value | $I^2$, % | z | *P*-value | t | *P*-value |
| *ApaI* | | | | | | | | | | | | | | |
| | a vs A | Overall | 5 | 0.98 | 0.83, 1.16 | 0.838 | F | 5.71 | 0.222 | 30.0 | 0.73 | 0.462 | 0.45 | 0.682 |
| | | Asian | 3 | 1.10 | 0.81, 1.48 | 0.550 | F | 1.05 | 0.592 | 0.0 | | | | |
| | | Caucasian | 2 | 0.90 | 0.60, 1.35 | 0.606 | R | 3.95 | 0.047 | 74.7 | | | | |
| | aa + Aa vs AA | Overall | 5 | 1.38 | 1.07, 1.79 | 0.014 | F | 3.72 | 0.445 | 0.0 | 0.24 | 0.806 | 0.42 | 0.703 |
| | | Asian | 3 | 1.56 | 0.95, 2.56 | 0.082 | F | 2.50 | 0.287 | 19.9 | | | | |
| | | Caucasian | 2 | 1.32 | 0.98, 1.79 | 0.069 | F | 0.93 | 0.335 | 0.0 | | | | |
| | aa vs Aa + AA | Overall | 5 | 1.05 | 0.80, 1.39 | 0.721 | F | 3.35 | 0.501 | 0.0 | 0.24 | 0.806 | 0.28 | 0.799 |
| | | Asian | 2 | 0.86 | 0.53, 1.38 | 0.520 | F | 1.19 | 0.551 | 0.0 | | | | |
| | | Caucasian | 2 | 1.17 | 0.83, 1.64 | 0.370 | F | 1.07 | 0.301 | 6.4 | | | | |
| | aa vs AA | Overall | 5 | 1.40 | 1.00, 1.96 | 0.052 | F | 1.76 | 0.780 | 0.0 | 0.24 | 0.806 | 0.62 | 0.581 |
| | | Asian | 3 | 1.45 | 0.75, 2.82 | 0.268 | F | 0.19 | 0.909 | 0.0 | | | | |
| | | Caucasian | 2 | 1.38 | 0.93, 2.04 | 0.109 | F | 1.55 | 0.213 | 35.4 | | | | |
| | Aa vs AA | Overall | 5 | 1.48 | 1.12, 1.95 | 0.006 | F | 2.77 | 0.597 | 0.0 | 1.22 | 0.221 | 3.03 | 0.056 |
| | | Asian | 3 | 2.06 | 1.21, 3.52 | 0.008 | F | 0.34 | 0.842 | 0.0 | | | | |
| | | Caucasian | 2 | 1.31 | 0.95, 1.81 | 0.106 | F | 0.41 | 0.524 | 0.0 | | | | |
| *BsmI* | | | | | | | | | | | | | | |
| | b vs B | Overall | 5 | 1.02 | 0.80, 1.28 | 0.604 | R | 11.08 | 0.026 | 63.9 | −0.24 | 1.000 | −0.09 | 0.937 |
| | | Asian | 2 | 0.90 | 0.72, 1.11 | 0.308 | F | 0.94 | 0.333 | 0.0 | | | | |
| | | Caucasian | 3 | 1.10 | 0.76, 1.61 | 0.604 | R | 7.78 | 0.020 | 74.3 | | | | |
| | bb + Bb vs BB | Overall | 5 | 1.03 | 0.84, 1.26 | 0.777 | F | 6.99 | 0.136 | 42.8 | 0.73 | 0.462 | −1.33 | 0.277 |
| | | Asian | 2 | 1.26 | 0.96, 1.66 | 0.101 | F | 0.32 | 0.574 | 0.0 | | | | |
| | | Caucasian | 3 | 0.80 | 0.59, 1.09 | 0.162 | F | 2.11 | 0.349 | 5.1 | | | | |
| | bb vs Bb + BB | Overall | 5 | 0.81 | 0.64, 1.04 | 0.103 | F | 5.22 | 0.266 | 23.3 | 0.24 | 0.806 | −0.95 | 0.413 |
| | | Asian | 2 | 0.90 | 0.54, 1.50 | 0.681 | F | 0.11 | 0.738 | 0.0 | | | | |
| | | Caucasian | 3 | 0.72 | 0.45, 1.17 | 0.184 | R | 4.92 | 0.085 | 59.3 | | | | |
| | bb vs BB | Overall | 5 | 0.72 | 0.56, 0.99 | 0.042 | F | 5.11 | 0.276 | 21.7 | 0.24 | 0.806 | −0.33 | 0.765 |
| | | Asian | 2 | 0.80 | 0.43, 1.49 | 0.489 | F | 0.00 | 0.985 | 0.0 | | | | |
| | | Caucasian | 3 | 0.66 | 0.36, 1.20 | 0.176 | R | 4.96 | 0.084 | 59.7 | | | | |
| | Bb vs BB | Overall | 5 | 1.11 | 0.89, 1.37 | 0.361 | F | 4.02 | 0.404 | 0.4 | 0.73 | 0.462 | −1.93 | 0.149 |
| | | Asian | 2 | 1.30 | 0.98, 1.74 | 0.070 | F | 0.45 | 0.504 | 0.0 | | | | |
| | | Caucasian | 3 | 0.89 | 0.64, 1.24 | 0.494 | F | 0.65 | 0.724 | 0.0 | | | | |

*(continued on next page)*

Guo et al. (2023), *PeerJ*, DOI 10.7717/peerj.15181

**Table 3** (*continued*)

| SNP | Comparison | Subgroup | No. of studies | Test of association | | | Tests of heterogeneity | | | | Begg test for publication test | | Egger test for publication bias | |
|---|---|---|---|---|---|---|---|---|---|---|---|---|---|---|
| | | | | OR | 95% CI | *P*-value | Model | Q | *P*-value | $I^2$, % | z | *P*-value | t | *P*-value |
| *FokI* | | | | | | | | | | | | | | |
| | f vs F | Overall | 8 | 1.08 | 0.88, 1.34 | 0.459 | R | 28.55 | <0.001 | 75.5 | 0.12 | 0.902 | 0.52 | 0.623 |
| | | Asian | 5 | 1.08 | 0.78, 1.50 | 0.631 | R | 25.90 | <0.001 | 84.6 | | | | |
| | | Caucasian | 3 | 1.13 | 0.94, 1.36 | 0.177 | F | 1.65 | 0.438 | 0.0 | | | | |
| | ff + Ff vs FF | Overall | 8 | 0.91 | 0.67, 1.23 | 0.531 | R | 15.33 | 0.032 | 54.3 | 0.12 | 0.902 | −0.53 | 0.613 |
| | | Asian | 5 | 0.85 | 0.48, 1.51 | 0.579 | R | 13.46 | 0.009 | 70.3 | | | | |
| | | Caucasian | 3 | 1.04 | 0.80, 1.35 | 0.795 | F | 1.23 | 0.541 | 0.0 | | | | |
| | ff vs Ff + FF | Overall | 8 | 1.23 | 0.88, 1.73 | 0.228 | R | 20.14 | 0.005 | 65.2 | 1.11 | 0.266 | −1.44 | 0.201 |
| | | Asian | 5 | 1.12 | 0.66, 1.91 | 0.671 | R | 18.06 | 0.001 | 77.9 | | | | |
| | | Caucasian | 3 | 1.43 | 1.01, 2.03 | 0.041 | F | 2.01 | 0.366 | 0.6 | | | | |
| | ff vs FF | Overall | 8 | 1.11 | 0.73, 1.70 | 0.615 | R | 17.00 | 0.017 | 58.8 | 0.87 | 0.386 | −1.24 | 0.262 |
| | | Asian | 5 | 0.98 | 0.48, 2.02 | 0.957 | R | 14.12 | 0.007 | 71.7 | | | | |
| | | Caucasian | 3 | 1.35 | 0.91, 2.01 | 0.130 | F | 2.71 | 0.258 | 26.1 | | | | |
| | Ff vs FF | Overall | 8 | 0.86 | 0.71, 1.04 | 0.127 | F | 9.08 | 0.247 | 22.9 | 0.12 | 0.902 | −0.60 | 0.573 |
| | | Asian | 5 | 0.79 | 0.60, 1.04 | 0.087 | F | 7.45 | 0.114 | 46.3 | | | | |
| | | Caucasian | 3 | 0.94 | 0.71, 1.24 | 0.659 | F | 0.85 | 0.655 | 0.0 | | | | |
| *TaqI* | t vs T | Overall | 3 | 1.18 | 0.93, 1.49 | 0.167 | F | 2.39 | 0.303 | 16.3 | 0.00 | 1.000 | −0.46 | 0.724 |
| | | Asian | 2 | 0.99 | 0.71, 1.37 | 0.933 | F | 0.06 | 0.802 | 0.0 | | | | |
| | | Caucasian | 1 | 1.42 | 1.02, 1.98 | 0.040 | | | | | | | | |
| | tt + Tt vs TT | Overall | 3 | 0.85 | 0.62, 1.16 | 0.296 | F | 2.85 | 0.240 | 29.9 | 0.00 | 1.000 | −0.05 | 0.969 |
| | | Asian | 2 | 1.06 | 0.71, 1.60 | 0.773 | F | 0.16 | 0.688 | 0.0 | | | | |
| | | Caucasian | 1 | 0.63 | 0.39, 1.01 | 0.054 | | | | | | | | |
| | tt vs Tt + TT | Overall | 3 | 0.78 | 0.50, 1.22 | 0.270 | F | 0.55 | 0.759 | 0.0 | 0.00 | 1.000 | 0.13 | 0.916 |
| | | Asian | 2 | 0.91 | 0.49, 1.69 | 0.769 | F | 0.00 | 0.963 | 0.0 | | | | |
| | | Caucasian | 1 | 0.65 | 0.34, 1.25 | 0.195 | | | | | | | | |
| | tt vs TT | Overall | 3 | 0.71 | 0.44, 1.13 | 0.145 | F | 1.31 | 0.520 | 0.0 | 0.00 | 1.000 | 0.07 | 0.955 |
| | | Asian | 2 | 0.90 | 0.48, 1.70 | 0.750 | F | 0.00 | 0.962 | 0.0 | | | | |
| | | Caucasian | 1 | 0.52 | 0.26, 1.05 | 0.068 | | | | | | | | |
| | Tt vs TT | Overall | 3 | 0.87 | 0.62, 1.20 | 0.387 | F | 2.11 | 0.348 | 5.4 | 0.00 | 1.000 | −0.08 | 0.952 |
| | | Asian | 2 | 1.06 | 0.69, 1.63 | 0.798 | F | 0.16 | 0.691 | 0.0 | | | | |
| | | Caucasian | 1 | 0.66 | 0.40, 1.09 | 0.104 | | | | | | | | |

**Notes.**

VDR, Vitamin D receptor; HDP, Hypertensive Disorders of Pregnancy; SNP, single nucleotide polymorphism; OR, odds ratio; 95% CI, 95% confidence interval; F, fixed effect model; R, random effect model.

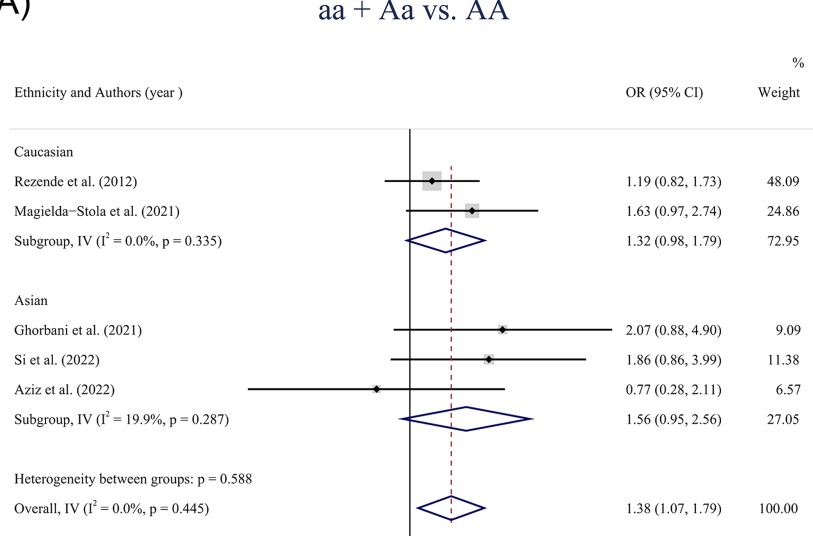

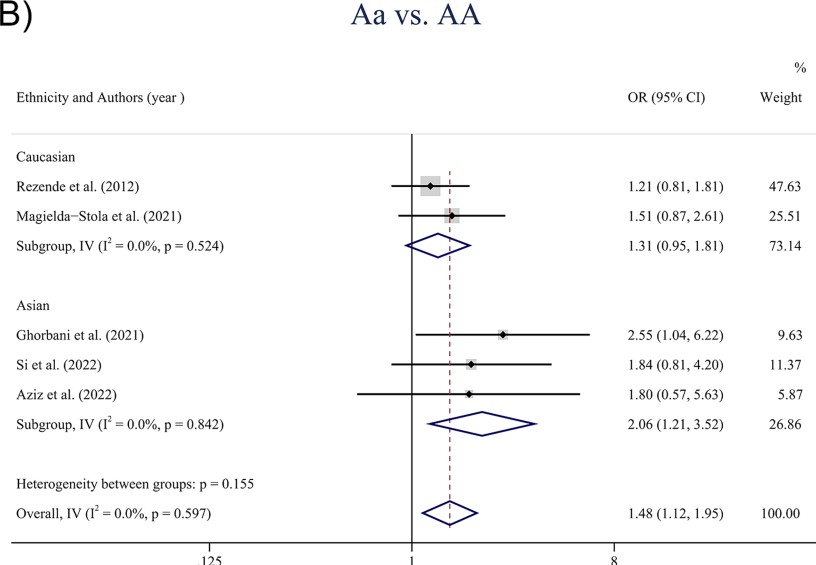

**Figure 2** **Forest plot for pooled odds ratio (OR) and the corresponding 95% confidence interval (CI) of the association between the *ApaI* polymorphism and hypertensive disorders of pregnancy (HDP).** (A) Dominant model (aa + Aa *vs.* AA); (B) heterozygote model (Aa *vs.* AA).

a 48% increased risk of HDP compared with AA carriers, and a 2.06-fold increased risk was observed in Asians. However, no association between the *ApaI* polymorphism and the risk of HDP was observed among Caucasians in the subgroup analysis. This study also found the *VDR* gene *BsmI* polymorphism had an association with HDP susceptibility in the homozygote model. The *BsmI* bb variant provided 28% more protection against HDP compared with the BB genotype. Besides, the association between the *VDR* gene *FokI*
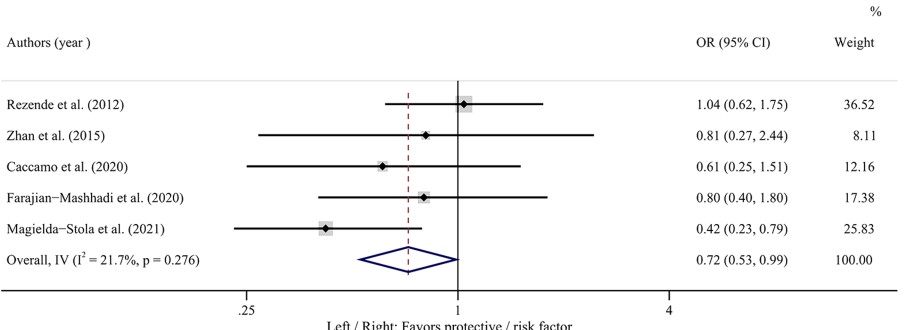

**bb vs. BB**

| Authors (year) | | OR (95% CI) | % Weight |
|---|---|---|---|
| Rezende et al. (2012) | | 1.04 (0.62, 1.75) | 36.52 |
| Zhan et al. (2015) | | 0.81 (0.27, 2.44) | 8.11 |
| Caccamo et al. (2020) | | 0.61 (0.25, 1.51) | 12.16 |
| Farajian−Mashhadi et al. (2020) | | 0.80 (0.40, 1.80) | 17.38 |
| Magielda−Stola et al. (2021) | | 0.42 (0.23, 0.79) | 25.83 |
| Overall, IV ($I^2$ = 21.7%, p = 0.276) | | 0.72 (0.53, 0.99) | 100.00 |

Left / Right: Favors protective / risk factor

**Figure 3  Forest plot for pooled odds ratio (OR) and the corresponding 95% confidence interval (CI) of the association between the *BsmI* polymorphism and hypertensive disorders of pregnancy (HDP) in the homozygote model (bb *vs.* BB).**

polymorphism and HDP was only found statistically significant in Caucasians, but not in the overall population. This may be due to a single study that reported a relatively stronger association, rather than a common high frequency of susceptible genotype in the Caucasian population. For the *VDR* gene *TaqI* polymorphism, only one study reported a statistically significant association in the Caucasian population. Thus, the results of our current study still cannot sufficiently clarify the role of the *VDR* gene *FokI* and *TaqI* polymorphisms in the occurrence of HDP, and the positive findings observed should be only considered exploratory, and future studies with larger sample sizes still need to confirm these findings.

The following points are worth noting when interpreting our integrated findings. First, differences in ethnicity may contribute to the variability in our findings on the relationship between the *VDR* gene *ApaI* polymorphism and HDP. The *VDR* gene is highly polymorphic, and the frequencies of its alleles were highly variable among different ethnicities (*Valdivielso & Fernandez, 2006*). Thus, the VDR affinity for vitamin D metabolites may also vary by ethnicity, which alters individual susceptibility to 1,25-(OH)$_2$ D$_3$ (*Haussler et al., 1998*). In this sense, our results can be supported by previous studies, *e.g.*, *Ghorbani et al. (2021)* reported that the *ApaI* (G>T) GT variant was associated with preeclampsia in Iran pregnant women (GT *vs.* GG: OR: 2.55; 95% CI [1.04–6.22]; $P = 0.04$), while another study conducted among the Polish did not found such association in the heterozygous model (OR: 1.51; 95% CI [0.87–2.61]) (*Magiełda-Stola et al., 2021*). Besides, this explanation can be supported by previous studies on the concentrations of vitamin D. One study conducted in Egypt reported women carrying mutant alleles for the *ApaI* polymorphism showed significantly lower serum 25-(OH) D$_3$ levels than those with the wild genotypes (aa + Aa *vs.* AA:13.5 ± 1.4 *vs.* 17.4 ± 1.5; $P < 0.05$) (*Zaki et al., 2017*), while another study indicated the *ApaI* (C>A) CA variant was not correlated with maternal 25-hydroxyvitamin D$_3$ (25-(OH) D$_3$) levels ($\beta = -2.65$; 95% CI [−10.83–5.51]; $P = 0.52$) for women in Brazil (*Pereira-Santos et al., 2019*). However, the insufficient number of current studies included could not rule out the possibility of

(A)

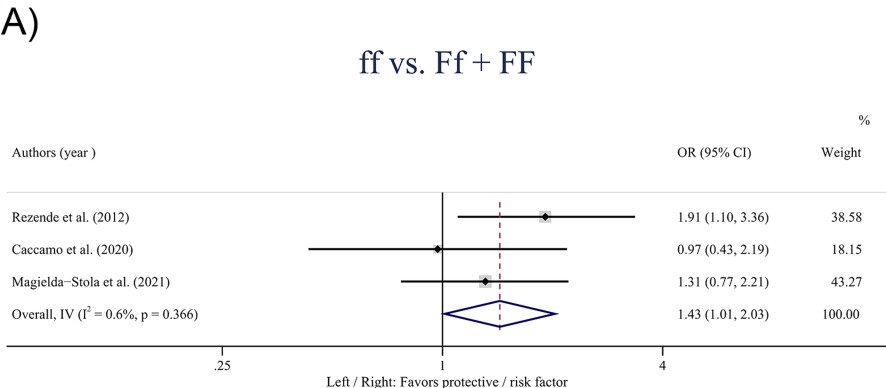

(B)

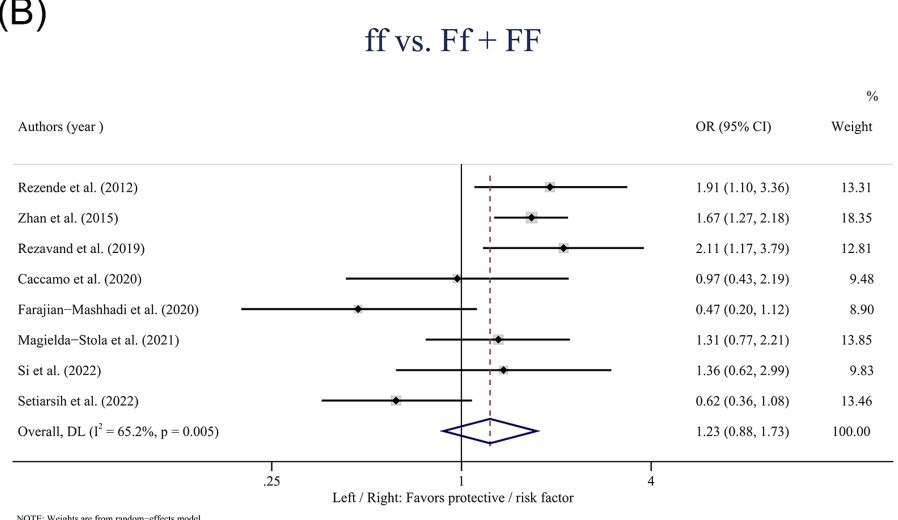

**Figure 4** **Forest plot for pooled odds ratio (OR) and the corresponding 95% confidence interval (CI) of the association between the *FokI* polymorphism and hypertensive disorders of pregnancy (HDP).** (A) Recessive model (ff + Ff *vs.* FF) in Caucasians; (B) Recessive model (ff *vs.* Ff + FF) in the overall population.

sampling error and publication bias, which can also affect the results. Second, since the *FokI* polymorphism has consequences for both VDR protein structure and transcriptional activity (*Whitfield et al., 2001*), most studies have examined the association of the *VDR* gene *FokI* polymorphism with HDP susceptibility. Our meta-analysis failed to provide adequate evidence to support the association between the *FokI* polymorphism and the risk of HDP. This finding is consistent with most prior studies (*Caccamo et al., 2020*; *Magiełda-Stola et al., 2021*; *Rezavand et al., 2019*; *Rezende et al., 2012*; *Si et al., 2022*), while there were also studies that had contrary results (*Farajian-Mashhadi et al., 2020*; *Setiarsih, Hastuti & Nurdiati, 2022*; *Zhan et al., 2015*), *e.g.*, *Farajian-Mashhadi et al. (2020)* and *Zhan et al. (2015)* reported the f allele of the *FokI* polymorphism as the protective factor and risk factor for HDP, respectively. On the other hand, given the potential mediating role

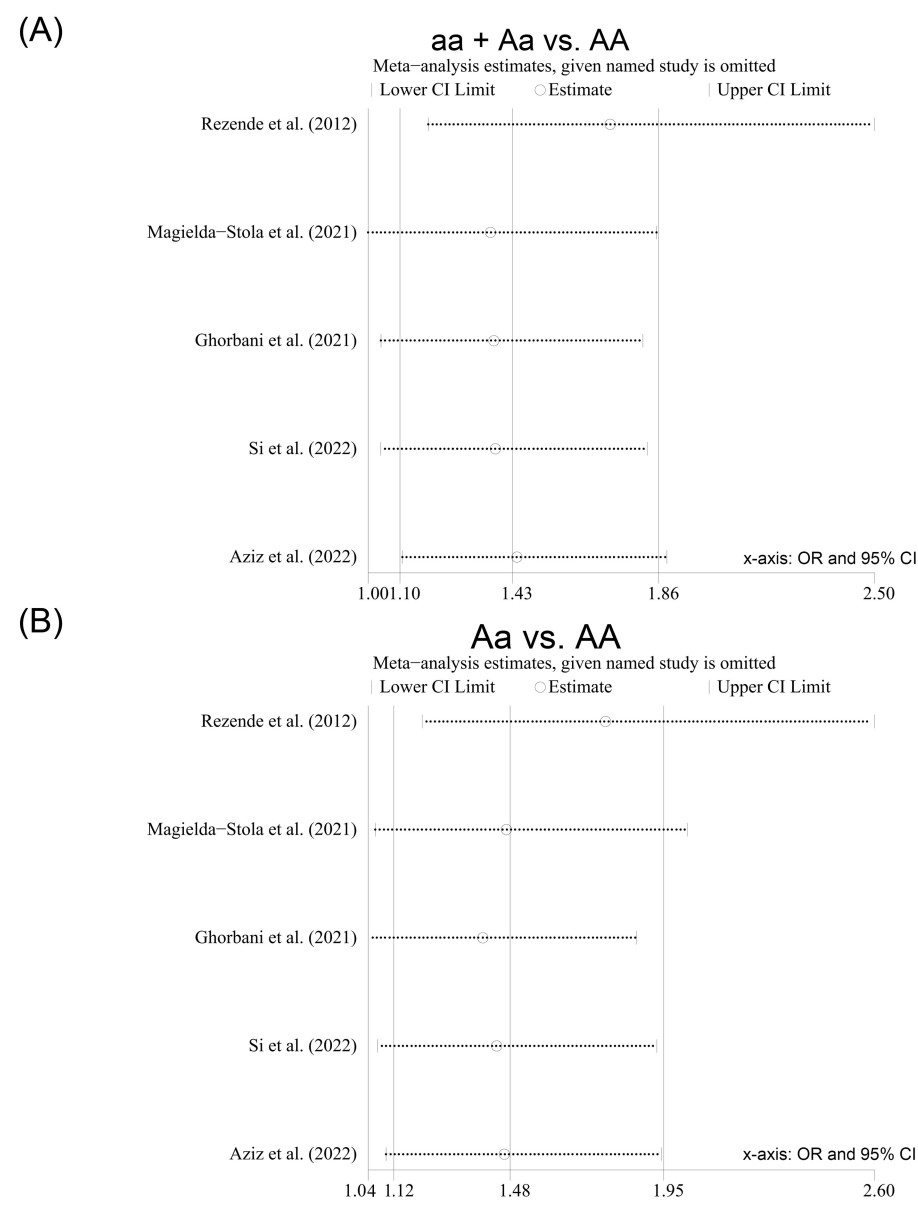

**Figure 5** **Sensitivity analysis of the studies included for the *ApaI* polymorphism; (A) Dominant model (aa + Aa *vs.* AA); (B) heterozygote model (Aa *vs.* AA).**

of vitamin D status in this association (*Caccamo et al., 2020*), the exact role played by the *VDR* gene *FokI* polymorphism in vitamin D concentrations remains obscure as well. *Monticielo et al. (2012)* reported significantly higher concentrations of 1,25-$(OH)_2D_3$ in Brazil subjects with the *FokI* f/f genotype than those with the F/F genotype ($31.6 \pm 14.1$ ng/ml *vs.* $23.0 \pm 9.2$ ng/ml; $P = 0.004$). On the contrary, another study conducted by Karras et al. in Greece suggested that mothers with the *FokI* F/F polymorphism had a 70% lower risk of vitamin D deficiency compared with f/f ones (OR: 0.30; 95% CI [0.09–0.92]; $P = 0.03$) (*Karras et al., 2020*), however, there was also a study revealing that the *FokI* f/f

genotype was not associated with vitamin D levels and deficiency of vitamin D among a Greek rural population (OR: 0.56; 95% CI [0.29–1.10] $P = 0.09$) (*Divanoglou et al., 2021*). Third, the heterogeneity did not decrease in parallel after subgroup analysis based on ethnic groups, indicating that the inconsistent results of current studies and the heterogeneity of this meta-analysis might not only be derived from the differences in the sample sizes, populations, or ethnicities of the subjects, but gene-environment interactions that need to be considered. One study conducted by *Serrano et al. (2020)* preliminarily displayed the interaction between alcohol consumption and family history in preeclampsia patients, but the available evidence is absent for the existence of interaction effects between the *VDR* gene SNPs and environmental risk factors on HDP. Further studies are needed to clarify the complex gene-gene, gene-environment, and gene-nutrient interactions.

Although the mechanisms through how the *VDR* gene polymorphisms affect the risk of HDP are still not entirely clear, it is rational in biology. Evidence for the association of the *VDR* gene polymorphisms with common risk factors for HDP has been reported in previous studies, such as obesity (*Chen et al., 2019*), GDM (*Zeng et al., 2022*), hypertension susceptibility (*Zhu et al., 2019*), chronic kidney disease (CKD) susceptibility (*Santoro et al., 2015*), etc. Furthermore, the *VDR* gene polymorphisms might be involved in target organ damage in hypertensive patients (*Kulah et al., 2006*). On the other hand, vitamin D deficiency was found associated with endothelial dysfunction and vascular damage (*Kim et al., 2020*). Vitamin D has been proven to downregulate the renin-angiotensin-aldosterone system (RAAS), which is one of the essential mechanisms of blood pressure regulation (*Giménez et al., 2020*). Since the VDR is extensively expressed in cardiomyocytes and vascular endothelial cells, and the $1,25\text{-}(OH)_2D_3$ may suppress the RAAS to maintain stable BP by binding to the VDR (*Giménez et al., 2020*). Based on that, vitamin D supplementation during pregnancy was regarded to be protective against preeclampsia (*Khaing et al., 2017*), and the response to vitamin D supplementation can also be regulated by the *VDR* gene (*Usategui-Martín et al., 2022*). In general, our findings provided clues for future research on the pathogenesis of HDP and might have clinical implications. Obstetricians may better stratify the risk of HDP and develop appropriate prevention strategies and personalized treatments by considering maternal genotype in the clinical work.

The present meta-analysis has several limitations that should be considered. First, the number of eligible studies included in this meta-analysis was relatively small. This limited the strength of evidence for our findings and further investigation in the meta-analysis. We did not conduct the subgroup analysis based on the subtypes of HDP since the grouping status for gestational hypertension and preeclampsia was not available in most studies included. In addition, data provided by current studies on stratification by ethnicity was limited, thus constraining our further elucidation of ethnic differences. Second, the existence of potential confounding factors could not be ruled out, including obesity, smoking, alcohol intake, *etc.*, and these possible confounding factors might bias the results of our meta-analysis when pooling the unadjusted results. Third, an obvious heterogeneity was observed among these studies, indicating that the results from current studies are still characterized by considerable uncertainty and controversy and the pooled results should be interpreted with caution.

## CONCLUSIONS

In conclusion, our current meta-analysis provides evidence that the *VDR* gene *ApaI* and *BsmI* polymorphisms may be associated with the susceptibility risk of HDP. The existing evidence is insufficient to conclude that there are ethnic differences in the association of the *VDR* gene polymorphisms with HDP. Therefore, more case-control studies of high quality with larger sample sizes from multiple ethnic groups deserve to be launched to further confirm our findings.

**Abbreviations**

| | |
|---|---|
| **VDR** | vitamin D receptor |
| **HDP** | hypertensive disorders of pregnancy |
| **OR** | odds ratios |
| **CI** | confidence intervals |
| **BP** | blood pressure |
| **GDM** | gestational diabetes mellitus |
| **ACE** | angiotensin-converting enzyme |
| **MTHFR** | methylenetetrahydrofolate reductase |
| **TNF-** | tumor necrosis factor-$\alpha$ |
| **COMT** | catechol-O-methyltransferase |
| **AGT** | angiotensinogen |
| **eNOS** | endothelial nitric oxide synthase |
| **RAAS** | renin-angiotensin-aldosterone system |
| **1,25-(OH)2D3** | 1,25-Dihydroxyvitamin D3 |
| **RXR** | unoccupied retinoid X receptor |
| **VDREs** | vitamin D responsive elements |
| **SNP** | single nucleotide polymorphism |
| **PRISMA** | Preferred Reporting Items for Systematic Reviews and Meta-Analyses |
| **PROSPERO** | International Prospective Register of Systematic Reviews |
| **HWE** | Hardy-Weinberg equilibrium |
| **NOS** | Newcastle-Ottawa Scale |
| **GH** | gestational hypertension |
| **PE** | pre-eclampsia |
| **GWAS** | Genome-Wide Association Studies |
| **MR** | Mendelian Randomization |
| **PCR-RFLP** | polymerase chain reaction-restriction fragment length polymorphism; |
| **TaqMan qPCR** | TaqMan-Based real-time polymerase chain reaction |
| **MALDI-TOF MS** | matrix-assisted laser desorption ionization time-of-flight mass |
| **PCR** | spectrometry coupled with single-base extension polymerase chain reaction |
| **AS-PCR** | allele-specific polymerase chain reaction |

### Funding
The authors received no funding for this work.

### Competing Interests
The authors declare there are no competing interests.

### Author Contributions
- Yicong Guo conceived and designed the experiments, performed the experiments, analyzed the data, prepared figures and/or tables, authored or reviewed drafts of the article, and approved the final draft.
- Yu Zhang conceived and designed the experiments, performed the experiments, analyzed the data, prepared figures and/or tables, authored or reviewed drafts of the article, and approved the final draft.
- Xiangling Tang performed the experiments, prepared figures and/or tables, and approved the final draft.
- Xionghao Liu conceived and designed the experiments, authored or reviewed drafts of the article, and approved the final draft.
- Huilan Xu conceived and designed the experiments, authored or reviewed drafts of the article, and approved the final draft.

### Data Availability
The data described in this article is available from the original published articles in Tables 1 and 3.

### Supplemental Information
Supplemental information for this article can be found online at http://dx.doi.org/10.7717/peerj.15181#supplemental-information.

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
