# Peer review of "Association between Vitamin D receptor (VDR) gene polymorphisms and hypertensive disorders of pregnancy: a systematic review and meta-analysis"

_PeerJ, doi:10.7717/peerj.15181_

## Round 0.1 · original submission · Major Revisions

The manuscript has been assessed by three independent reviewers and I strongly suggest addressing the concerns raised by all three reviewers before your paper can be considered for publication. I appreciate the authors for including limitations in the discussion section. The authors are suggested to modify the inclusion criteria to include studies from other ethnic groups and re-analyze. It is also recommended to discuss the clinical implications of the current study.

Reviewer 1 ·

Basic reporting

This paper is overall well-written, with a clear introduction and background. The result part is also well-structured with good data presentations. For figures, I wish the authors could add x-axis labels to the existing figures to make it more clear for readers, e.g. figure 2 to 5.

Experimental design

This study sought to perform a meta-analysis of VDR gene variation and HDP, which is timely, well-defined, and meaningful analysis. And this study certainly fits the aim and scope of this journal.
The technical aspects of this analysis are also good, with a clear-articulated method part.

Validity of the findings

The major findings and conclusions of this study are supported by the author's analyses. Statistics are sound and well-controlled. I do not have concern regarding the analyses and conclusions.

Additional comments

Some minor points about references need to be addressed:
1. line 58, reference is incomplete.
2. line 62, reference is incomplete.
3. line 81, reference is incomplete.

·

Basic reporting

Appreciate the efforts taken by authors to revise the manuscript according to the comments given by reviewers in BMC journal. The manuscript is well written. The use of NOS method for scoring the articles and new prisma guidelines for organizing the review are appreciable.

The main drawback i could see in this study is the number of studies included for meta analysis and the results obtained

This study doesn't help to attain a conclusive evidence on type of VDR gene polymorphism and its relation with HDP. The authors have clearly described in their conclusion that this study does not help to infer any concrete information based on the meta analysis.

I suggest to repeat the analysis with inclusion of one or two relevant clinical conditions (DM or Obesity) or other forms of hypertension disorder and their association with VDR gene polymorphism.

In discussion i could see that authors have included VDR gene polymorphism results of other ethnic groups. It is good to modify the inclusion criteria to include studies from other ethnic groups as well. By this way convincing results can be derived.

Experimental design

Authors have followed all necessary guidelines for reporting of Systematic reviews and meta-analysis.

The research questions are well defined by authors. However the outcome of this study doesn't help to get a strong evidence of association between VDR gene polymorphism and HDP. The use of statistical methods for analysis are described well.

Validity of the findings

The study is unique. However it doesn't provide clear solution to the research questions that this study is based upon. The use of statistical methods are appropriate.

When convincing results are not obtained from meta analysis why did the authors attempt to publish this results? Please justify this. Whether this study address the research gaps in this field? what will be the future directions that one can obtain from this study?

Please try to modify the study objectives if you didn't get valid results..

Additional comments

Please refer to the attached PDF file for specific comments. The quality of figures has to be improved

Reviewer 3 ·

Basic reporting

The manuscript on, systematic review and meta-analysis on identifying the association between Vitamin D receptor (VDR) gene polymorphisms and hypertensive disorders caused during pregnancy is an interesting study. The manuscript is well-written and provides a thorough introduction to the scientific problem by referencing relevant literature appropriately. The presentation of results is clear and the authors have followed the PRISMA guidelines precisely in presenting their work.

Experimental design

Overall, the study design and execution are well done. However, I question if the authors used very strict inclusion criteria or performed subgroup analysis, which might have led to insufficient evidence or inconclusive results.

Validity of the findings

The article is well-written and presents the limitations of the study, however, the inconclusive results raise many questions and are open to different interpretations.

Did the authors validate the effectiveness of the keyword selection? With only 173 articles, this may result in a limited number of studies for inclusion in the meta-analysis and may lead to insufficient evidence.

It is important to discuss the potential benefits of this study at the end.

what additional analyses or research designs authors can plan to use to further explore their findings?

Additional comments

The authors presented keywords used to search for literature at three different places in different ways. It would be advisable not to use the keywords “VDR gene” and “HDP” in the abstract which will be misleading the reader.

---

## Round 0.2 · Minor Revisions

Authors have adequately addressed the comments made by the reviewers in the revised manuscript. However, the manuscript will be ready for publication after a minor revision on the presentation of data as suggested by one of the reviewers.

Reviewer 1 ·

Basic reporting

The authors have addressed my previous questions

Experimental design

The authors have addressed my previous questions

Validity of the findings

The authors have addressed my previous questions

·

Basic reporting

1. The authors have made efforts to address many of my comments, which is appreciable. The manuscript is structured well. Only minor comments are not addressed which can be corrected by the authors.

2. I feel that it is good to include funnel and Egger test results as a supplementary file. I agree that the publication bias is less when limited studies are used. However, not all readers can appreciate this, unless the results are provided for any one plot of your convenience as a supplementary file.

3. In figure 1, please try to include the different exons, exon boundaries boundaries and the location of SNP's.

4. Please confirm whether the figure 3 and 4 has included the results of the additional study along with the previously computed results. You have mentioned that new study was included after our suggestion. I raised this question due to the difficulty in identification of the 9 studies included for analysis. I was not sure which one was the recently added study. According to me, it doesn't take more time to code the nine studies and modify the changes in the manuscript and figure. Anyways I leave this comment to the decision of editor and other reviewers to decide if the suggested correction is appropriate.

Why the I2 value is zero? Does it mean that there is no heterogeneity seen between the study factors? The reason for this has to be mentioned in the manuscript.

Experimental design

Looks good. This manuscript falls within the scope of this journal and i am sure it will reach wider audience, especially to researchers working in Hypertension and pregnancy.

Validity of the findings

The study is novel. Only minor corrections are required in the representation of data.

Additional comments

No specific comments.

Reviewer 3 ·

Basic reporting

no comment

Experimental design

no comment

Validity of the findings

no comment

Additional comments

Authors have adequately addressed the comments made by the reviewers in the revised version of the manuscript.

---

## Round 0.3 · accepted · Accept

The authors have addressed all the comments adequately and the manuscript is ready for publication.